# Loneliness and delusion-like experiences among women: Mediating role of procrastination and boredom proneness

Karolina Szalińska [ID]*

Institute of Psychology, Department of Social Sciences, University of Szczecin, Szczecin, Poland

* karolina.szalinska@usz.edu.pl

## Abstract

Delusions, a core psychopathological symptom, occur both in mental disorders and as delusion-like experiences in the general population. This study aimed to examine the relationship between loneliness and delusion-like experiences, considering the mediating roles of procrastination and boredom proneness. The sample consisted of 291 women aged 18–65 years ($M = 30.05$; $SD = 10.298$). The analyses revealed that procrastination ($Indirect = 0.05$; 95%$CI$ [0.01; 0.10] significantly mediated the relationship between loneliness and delusion-like experiences, while boredom proneness showed no significant mediating effect ($Indirect = 0.02$; 95%$CI$ [−0.01; 0.04]). These findings suggest that procrastination may be a key mechanism underlying the association between loneliness and delusions, with potential implications for therapeutic interventions. The results obtained may serve as a foundation for implementing effective interventions to reduce belief in delusions, stress associated with experiencing delusions, and preoccupation with delusional thoughts. Moreover, understanding the functioning of individuals with delusion-like experiences may support the adaptation of specific therapeutic and support techniques.

## Introduction

Delusions are psychopathological symptoms accompanying various mental disorders, such as bipolar disorder, paranoid schizophrenia, Alzheimer's disease, and dementia with Lewy bodies [1]. The Diagnostic and Statistical Manual of Mental Disorders [2] describes delusions as persistent beliefs that, despite contradictory evidence, are considered unquestionable [1,3–5]. These faulty beliefs can also occur to a lesser extent in healthy individuals, manifesting as delusion-like experiences that are less intense, less enduring, and more adaptive [6].

This concept is reflected in the psychosis continuum hypothesis, which highlights that the phenotype of psychotic disorders can range from nonclinical, non-pathological individual traits to clinical symptoms manifesting within mental illnesses

provided the original author and source are credited.

**Data availability statement:** The data underlying the results presented in the study are available from (https://osf.io/smky9/?view_only=339a37dac63a490eb54ccc0424c7962d)

**Funding:** The author(s) received no specific funding for this work.

**Competing interests:** The authors have declared that no competing interests exist.

[7]. A study conducted by Akcaoglu [8] in Belgium demonstrated that women experience psychotic-like experiences – including delusions – more frequently than men. Furthermore, the distress associated with these experiences was also higher among women. A key finding of the study was that the quality of social interactions was significantly related to symptom severity – women's distress linked to psychotic-like experiences appeared to be strongly influenced by the quality of their social relationships. Although delusions are most commonly studied in the context of mental disorders [9], increasing attention is being directed toward psychosocial factors that may explain their occurrence or intensification [10]. Among these factors, loneliness, procrastination, and boredom proneness play significant roles in human functioning, both in clinical and nonclinical populations [11–16]. Examining psychological mechanisms, such as procrastination and boredom proneness, in relation to loneliness and delusion-like experiences may help in better understanding individuals at the nonclinical end of the psychosis continuum.

*Hypothesis 1: Dimensions of loneliness positively correlate with the intensity of delusion-like experiences.*

Loneliness, understood as an emotional state, is an inherent human experience throughout the life cycle, particularly during early and late adulthood [17,18]. Loneliness results in a perceived lack of support and care from close ones, friends, family, or neighbors, and can even encompass political and economic exclusion. It may also be experienced as alienation from oneself [19]. Although loneliness is a universal experience, stemming from the need for belonging [20], and as a transient feeling, it is situationally determined, chronic loneliness appears maladaptive [21]. Barreto et al. [22], in a global study involving participants from 237 countries, found that levels of loneliness were higher among men than women. In the same study by Barreto et al. [22], a cross-country comparison based on cultural orientation revealed a decline in loneliness with age, but overall loneliness levels were higher in collectivistic societies than in individualistic ones. In contrast, a longitudinal study on loneliness conducted over a 15-year period revealed that women consistently reported higher levels of loneliness than men [23]. The authors suggest that these differences may stem from factors such as gender roles, societal expectations, and disparities in access to social support.

Loneliness, as a subjective experience of lacking satisfying social relationships, can provide a significant context for understanding mechanisms underlying delusion-like experiences.. Previous studies support the notion that loneliness is associated with paranoid thoughts [24]. McIntyre et al. [25] demonstrated a positive relationship between loneliness and paranoid beliefs among students. This finding was corroborated by [26] in both student and general population samples. Another study supports the idea that paranoid thoughts may arise as a result of loneliness in the general population [27]. Furthermore, Lamster et al. [28] documented that experimentally reducing loneliness is associated with a decrease in paranoid thoughts in nonclinical samples. Lin Toh et al.'s [29] study underscores the importance of examining loneliness in the context of delusions, indicating that loneliness significantly contributes to

delusion proneness. A reduced ability to self-regulate thoughts, behaviors, and emotions is one consequence of loneliness [17], which may be linked to a higher frequency of delusion-like experiences.

In light of prior findings indicating that women may be particularly vulnerable to the interaction between delusion-like experiences and loneliness, this study deliberately focused on female participants to explore these mechanisms with greater specificity.

### Hypothesis 2: Procrastination mediates the relationship between loneliness and delusion-like experiences.

Loneliness not only amplifies unpleasant emotional states but also impairs effective time management and decision-making, leading to phenomena such as procrastination [11,14,16], which is one of the mechanisms linking loneliness with delusion-like experiences. Theoretical underpinnings of this relationship can be found in emotion regulation models, such as the ego-depletion model [30], which posits that self-regulation capabilities are a finite resource. Chronic loneliness may deplete these resources, hindering effective action-taking and fostering task avoidance [31]. Procrastination may simultaneously exacerbate negative emotional states, leading to a spiral of deteriorating mental functioning [32].

Procrastination, understood as a chronic tendency to delay planned activities despite awareness of their negative consequences, may both result from and contribute to loneliness. Moreover, studies on procrastination have led Vodanovich and Rupp [33] to conclude that individuals who are prone to procrastination are more likely to experience boredom proneness. Their research revealed that procrastination is linked to aspects of boredom proneness, such as a lack of external stimulation, deficits in discovering internal interests, as well as emotional responses and perceptions of time. Further studies support the premise that boredom proneness is positively correlated with procrastination [34]. Individuals with high levels of boredom proneness exhibit significant difficulties focusing on tasks, redirecting their behavior toward alternative activities. A relationship has also been observed between boredom proneness and conspiracy-related paranoid thoughts [35]. These findings are explained by an individual's tendency to misinterpret aversive internal states—believing neutral events to be genuine problems. Furthermore, a sense of self-importance linked to boredom proneness may foster the belief that one possesses special knowledge concealed from the rest of the world.

### Hypothesis 3: Boredom proneness mediates the relationship between loneliness and delusion-like experiences.

Another significant mechanism concerning the variables discussed is boredom proneness, defined as a chronic tendency to experience boredom in situations that do not provide adequate stimulation [36]. In the context of psychopathology, boredom proneness may lead to misinterpreting neutral events as threatening, increasing the susceptibility to paranoid thoughts [35]. Cognitive models, such as the attention-engagement theory [37], suggest that individuals who are prone to boredom have difficulty maintaining focus on tasks, potentially amplifying negative emotional processing and contributing to delusion formation.

Cohen-Mansfield et al.'s [12] study offers crucial insights into delusions experienced by individuals with dementia in the context of boredom. The researchers observed that some delusions emerged during boredom, which they attributed to individuals adopting previous or imagined roles in the absence of a defined identity or role. Furthermore, this tendency primarily occurred when individuals were not engaged in any activity. Todman et al.'s [38] findings on the relationship between boredom and schizophrenia indicated that this clinical group attributes boredom to internal factors. Compared to students, individuals with schizophrenia were less likely to interpret boredom as being a result of social circumstances.

In summary, both procrastination and boredom proneness may represent key mechanisms explaining how loneliness translates into delusion-like experiences. The relatively limited number of studies on delusions and boredom proneness [12,35,38], the lack of sufficient empirical data focusing on the direct relationship between procrastination and delusion-like experiences, and the absence of data on their mediating roles in the relationship between delusion-like experiences and loneliness highlight the need to expand the information and data on the co-occurrence of these variables.

The primary objective of this study was to evaluate the relationships between loneliness and delusion-like experiences, with particular attention on the mediating roles of procrastination and boredom proneness. The findings may have practical implications for psychotherapy, offering insights into how to mitigate the potential effects of loneliness and procrastination on delusion formation. Additionally, the study may inform the design of psychological intervention programs aimed at addressing boredom and improving time management skills in individuals at increased risk of psychosis. This research also aimed to fill gaps in the existing literature by providing data on the psychological functioning of healthy individuals who experience delusion-like beliefs.

## Materials and methods

The chosen methodology facilitated the study's implementation through an electronic self-report format. The study was promoted via social media platforms, utilizing a purposive sampling approach. The research was approved by the Ethics Committee for Research Projects at the University of Szczecin (No. KB 47/2024). The study was conducted from November 22nd, 2024 to January 25th, 2025. The description of the questionnaire battery informed participants about the general purpose of the study, which was to assess beliefs and experiences related to the world, as well as its anonymous and voluntary nature. It was highlighted that the estimated completion time was 10–15 minutes. Participants responded to questions included in a demographic survey and four standardized instruments adapted for use in the Polish context. Before starting the study, each participant checked an informed consent to participate in the study and had read all information about the study.

*The Revised UCLA Loneliness Scale* (R-UCLA), developed by Russell et al. [39] and adapted into Polish by Kwiat-kowska [40], was used to measure loneliness. This scale consists of 20 items that form three dimensions. Participants responded on a 4-point scale ranging from 1 (*never*) to 4 (*often*). The subscales assess aspects such as (1) intimate others, referring to the feeling of exclusion; (2) social others, referring to the lack of closeness and support in relationships; and (3) belonging and affiliation, referring to the lack of community bonds. A total score reflecting overall loneliness can be obtained by summing the responses across all items. Higher scores on the R-UCLA indicate limited or absent interpersonal networks, feelings of rejection, and low social belonging. The Cronbach's alpha reliability coefficients were 0.94 ($CR = 0.91$) for the overall scale, 0.90 ($CR = 0.84$) for intimate others, 0.88 ($CR = 0.81$) for social others, and 0.77 ($CR = 0.63$) for belonging and affiliation.

*The Pure Procrastination Scale* (PPS), created by Steel [41] and adapted into Polish by Stępień and Cieciuch [42], contains 12 items assessing three dimensions of procrastination. Participants rated their responses on a 5-point scale ranging from 1 (*does not describe me at all*) to 5 (*describes me perfectly*). The tool comprises three subscales: decisional, behavioral, and maladaptive procrastination. An overall procrastination score is calculated by summing all items. The internal consistency, as measured by Cronbach's alpha, was high: 0.93 ($CR = 0.93$) for the total score, 0.88 ($CR = 0.77$) for decisional procrastination, 0.94 ($CR = 0.76$) for behavioral procrastination, and 0.80 ($CR = 0.88$) for maladaptive procrastination.

*The Boredom Proneness Scale* (BPS), originally developed by Farmer and Sundberg [43] and adapted into Polish by Flakus [44], was used to measure the susceptibility to boredom. Participants answered 12 items on a 7-point Likert scale ranging from *strongly disagree* to *strongly agree*. The scale comprises two subscales—internal stimulation and external stimulation—as well as an overall score. The Cronbach's alpha coefficient for the total score was 0.55 ($CR = 0.84$), with 0.57 ($CR = 0.73$) for internal stimulation and 0.72 ($CR = 0.73$) for external stimulation.

*The Peters et al. Delusions Inventory* (PDI) [45], developed to assess experiences similar to delusions, was adapted into Polish by Prochwicz and Gawęda [46]. The PDI includes 40 dichotomous questions (*yes/no*), and for each affirmative response, participants provide additional ratings on three dimensions: distress (1 = *not distressing at all* to 5 = *very distressing*), preoccupation (1 = *rarely think about it* to 5 = *think about it all the time*), and conviction (1 = *do not believe it is true at all* to 5 = *absolutely convinced it is true*). A total PDI score is calculated by summing the responses to all 40 primary

questions, while subscale scores are derived from the three dimensions. The Cronbach's alpha for the total score was 0.86 (*CR* = 0.78), and the reliability coefficients for distress, preoccupation, and conviction were 0.88 (*CR* = 0.82), 0.89 (*CR* = 0.82), and 0.87 (*CR* = 0.82), respectively.

The study sample comprised 291 women ranged in age from 18 to 65 years (*M* = 30.05, *SD* = 10.98). The sample varied in terms of educational attainment and place of residence. The largest proportion of participants reported having secondary education (*n* = 142; 48.8%) or higher education (*n* = 132; 45.4%). Regarding place of residence, the majority of participants were from cities with populations exceeding 200,000 (*n* = 108; 37.1%) (Table 1)

## Results

The statistical analyses were performed using *IBM SPSS Statistics 27* with the *PROCESS macro 5.0* [47], which facilitates bootstrap mediation analysis with a random sampling of 5000 iterations. A 95% confidence interval (*p* < 0.05) was adopted for the interpretation of the empirical results used to test the formulated research hypotheses. The study achieved an optimal sample size (*N* = 250) to detect significant mediation effects with a medium effect size (*f²* = 0.20) and the desired statistical power (β = 0.80) for a model with three predictors [48,49].

Before conducting the main analyses, all psychological variables were evaluated to verify the assumptions required for the selection of appropriate statistical tests. Extreme observations were removed based on boxplot analyses of all questionnaire results. Influential outliers in six mediation models were also identified and excluded using Cook's distance, Mahalanobis distance, and leverage values. Observations were excluded if they failed to meet two out of three distance criteria. No missing data were identified in the dataset.

### Loneliness and delusion-like experiences

To test the first hypothesis regarding the correlation between loneliness and the severity of delusion-like experiences, Pearson's r correlation coefficients were calculated. The results revealed statistically significant relationships (*p* < 0.001) between these variables (Table 2). Specifically, there was a statistically significant positive association between loneliness and delusion-like experiences (*r* = 0.30, *p* < 0.001). Moreover, the strongest correlations were observed between distress related to delusions and a sense of lack of connection with others (*r* = 0.44, *p* < 0.001), as well as between a lack

**Table 1. Sociodemographic characteristics of participants.**

| Demographic category | (*N* = 291) |
|---|---|
| **Age group** | |
| 18–22 years old | 92 (31.6%) |
| 23–35 years old | 113 (38.8%) |
| 36–55 years old | 76 (26.1%) |
| 56–65 years old | 10 (3.4%) |
| **Educational level** | |
| Primary | 11 (3.8%) |
| Secondary | 142 (48.8%) |
| Vocational | 6 (2.1%) |
| Higher | 132 (45.4%) |
| **Residence** | |
| Rural area | 61 (21.0%) |
| Urban area to 50,000 inhabitants | 57 (19.6%) |
| Urban area from 50,000–200,000 inhabitants | 65 (22.3%) |
| Urban area above 200,000 inhabitants | 108 (37.1%) |

**Table 2. Pearson's r coefficient values for the correlation between loneliness and delusion-like experiences.**

|  | Delusion-Like Experiences | Distress | Preoccupation | Conviction |
|---|---|---|---|---|
| **Loneliness** | 0.30*** | 0.38*** | 0.35*** | 0.34*** |
| Intimate others | 0.37*** | 0.44*** | 0.42*** | 0.42*** |
| Social others | 0.17** | 0.24*** | 0.22*** | 0.19** |
| Belonging and affiliation | 0.16** | 0.24*** | 0.21*** | 0.17** |

**p<0.01; ***p<0.001.

of interpersonal contact with preoccupation ($r=0.42$, $p<0.001$) and conviction ($r=0.42$, $p<0.001$) with delusion-related thoughts.

**Mediating role of procrastination and boredom proneness in the relationship between loneliness and delusion-like experiences**

The verification of the remaining hypotheses (H2, H3) was carried out by analyzing the mediating effect of procrastination and boredom proneness on delusion-like experiences explained by the intensity of loneliness (Fig 1).

The obtained results for the direct relationship between loneliness and delusion-like experiences ($\beta=0.30$; $p<0.001$; 95%CI [0.09; 0.20]) indicate a good fit of the model to the data ($F(1,288) = 29.47$; $p<0.001$), which explained over 9% of the variance in the intensity of delusion-like experiences ($R^2=0.09$). The results for individual variables of the model also indicated a good fit for models considering the individual effect of procrastination ($F(1,288) = 56.27$; $p<0.001$) and boredom proneness ($F(2,288) = 76.74$; $p<0.001$), as well as for both variables simultaneously ($F(3,288) = 16.08$; $p<0.001$), explaining 16%, 4%, and 14% of the variance, respectively.

An analysis of the regression coefficients of individual paths in the model showed an insignificant mediating effect of boredom proneness ($Indirect=0.02$; 95%CI [−0.01; 0.04]) due to a statistically insignificant relationship between this variable and loneliness ($\beta=0.08$; $p>0.05$; 95%CI [−0.03; 0.14]). However, the analysis of indirect effects revealed statistically significant mediating effects of procrastination ($Indirect=0.05$; 95%CI [0.01; 0.10]) and the simultaneous inclusion of procrastination and boredom proneness ($Indirect=0.01$; 95%CI [0.01; 0.02]). Moreover, the analyses indicate a statistically significant total effect ($Indirect=0.03$; 95%CI [0.01; 0.06]), as the confidence interval did not include zero.

The model demonstrates a partial mediation effect, as the initial strength of the relationship between loneliness and delusion-like experiences ($\beta=0.30$; $p<0.001$; 95%CI [0.09; 0.20]) decreased when the two mediating variables were

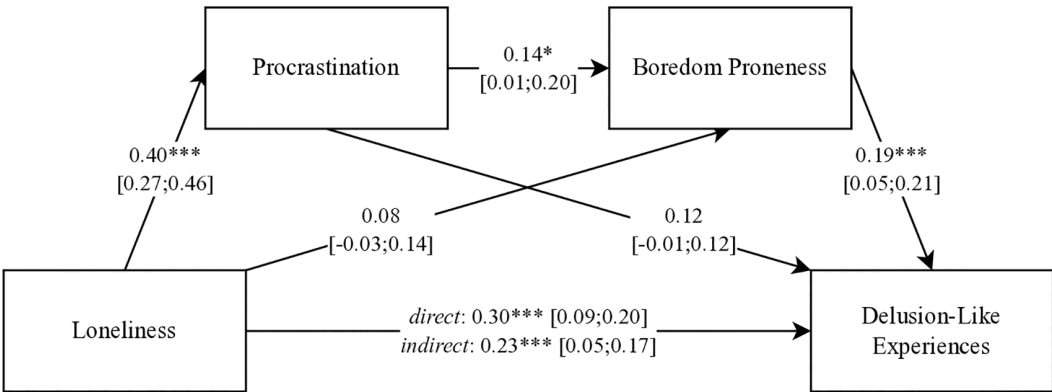

**Fig 1. Results of the analyzed mediation model (standardized weights).** * $p<0.05$; ** $p<0.05$; *** $p<0.001$.

included (β = 0.23; *p* < 0.001; 95%*CI* [0.05; 0.17]), while maintaining statistical significance (*p* < 0.001). In other words, procrastination and boredom proneness reduce the intensity of delusion-like experiences explained by loneliness, though procrastination appears to play a more substantial role.

## Discussion

This study aimed to evaluate the relationship between delusion-like experiences and loneliness, considering the mediating role of procrastination and boredom proneness among women. The analyses confirmed some findings from previous studies while providing new insights into the relationship between the examined variables.

A statistically significant positive correlation was found between loneliness and delusion-like experiences, as well as their components. This study aligns with analyses supporting the interdependence between these variables, particularly between loneliness and paranoid thoughts, which are a type of delusion and delusion-like experience [26,27]. Furthermore, increasing loneliness is associated with higher levels of psychotic-like experiences, including delusions, in the general population [50,51].

This study provides new insights into the relationship between loneliness and the dimensions of delusion-like experiences. The data indicate that all three components of this construct – distress, preoccupation, and conviction – are statistically significantly related to loneliness. The literature suggests [52] that satisfying social relationships yield positive outcomes, while difficulties in forming or maintaining relationships may lead to loneliness, anger, or anxiety. Such difficulties may also be linked to other issues, such as greater distress related to disturbing thoughts, higher levels of absorption in beliefs about the world, and stronger belief in false convictions due to a lack of feedback from the environment.

The analysis revealed that procrastination significantly mediates the examined relationship. This suggests that introducing this variable into the model weakens the relationship between loneliness and delusion-like experiences. The literature indicates that procrastination may be associated with cognitive processing difficulties [53], which are present in delusional experiences, suggesting a suppressive role of procrastination in the examined interdependence.

There is empirical evidence for the relationship between boredom and loneliness [54,55] as well as between boredom and delusions [12]. However, this study found that general boredom proneness, in conjunction with procrastination, mediates the relationship between loneliness and delusion-like experiences. The literature suggests that individuals with high external stimulation levels seek activities, interests, and excitement in the external environment, which may explain the role of these factors in the relationship between loneliness and delusion-like experiences [44]. In the study by Jaradat [56], women scored significantly higher than men on the external stimulation subscale. This suggests that women may be more susceptible to boredom due to low environmental stimulation, which could affect their engagement and motivation.

This study has certain limitations. The data were collected using an online survey distributed via social media platforms, which entails limited control over the research process and the potential for sampling bias. Nevertheless, this method is commonly used in psychological research and Internet findings seems to be consistent with findings from traditional methods [57]. The survey did not ask whether the participants had a diagnosed mental disorder or a family history of mental disorders. Additionally, the questionnaire battery did not include a tool that could identify individuals meeting diagnostic criteria, such as for delusional disorders (e.g., ICD-11 – 6A24). These data would have been significant in interpreting the results, as the psychometric tool used in this study applies to individuals both within the normative range of false beliefs and those at the extreme end of the continuum meeting diagnostic criteria. An important limitation of the present study was the relatively low Cronbach's alpha reliability coefficient for the BPS tool. This may result from the limited number of items in the scale, which significantly impacts the value of alpha [58], or from the characteristics of the sample, which may have lacked sufficient variability. Nevertheless, when a scale has a multidimensional structure, Cronbach's alpha represents only the lower bound of reliability [59], and previous studies [60,61] have reported similar difficulties with various versions of this instrument, indicating its structural instability. Despite the relatively low Cronbach's alpha values for the Boredom Proneness Scale, additional analyses were conducted to verify the instrument's internal consistency using

composite reliability (CR). CR accounts for individual factor loadings and measurement error, providing a more accurate estimation of reliability than Cronbach's alpha, which assumes equal loadings across items [62]. The results obtained showed acceptable CR values and indicate that, despite initial concerns based on alpha coefficients, the BPS demonstrates satisfactory internal consistency when assessed using a more robust metric. Thus, the inclusion of this tool in the model can be justified, although future studies should continue to explore and refine its psychometric properties.

Future research should include a balanced sample of men and women to explore individual differences in the relationship between delusions and sex. A longitudinal study would also help reveal changes in delusion-like experiences over time and identify partial cause-and-effect relationships. This could facilitate the development of strategies to support individuals experiencing delusions and loneliness, taking into account the role of boredom proneness and procrastination.

The study results have significant practical implications. Most importantly, they highlight the need to consider procrastination in interventions targeting individuals experiencing loneliness and delusion-like experiences. Techniques aimed at improving time management and self-regulation skills could help reduce the intensity of these experiences. Furthermore, the obtained data may contribute to a better understanding of the psychological mechanisms underlying these phenomena.

## Author contributions

**Conceptualization:** Karolina Szalińska.

**Data curation:** Karolina Szalińska.

**Formal analysis:** Karolina Szalińska.

**Funding acquisition:** Karolina Szalińska.

**Investigation:** Karolina Szalińska.

**Methodology:** Karolina Szalińska.

**Project administration:** Karolina Szalińska.

**Resources:** Karolina Szalińska.

**Software:** Karolina Szalińska.

**Supervision:** Karolina Szalińska.

**Validation:** Karolina Szalińska.

**Visualization:** Karolina Szalińska.

**Writing – original draft:** Karolina Szalińska.

**Writing – review & editing:** Karolina Szalińska.

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
