## [Decision Letter · Decision Letter 0]

PONE-D-25-06976Loneliness and delusion-like experiences: mediating role of procrastination and boredom pronenessPLOS ONE

Dear Dr. Szalińska,

Thank you for submitting your manuscript to PLOS ONE. After careful consideration, we feel that it has merit but does not fully meet PLOS ONE’s publication criteria as it currently stands. In particular, there are three issues. First, how data were collected, though many articles get published with a similar collection method. Second, it's mostly women, so no definitive generalization can be made. Third, an important Cronbach alpha is really low. There are two obvious ways to address (2) - either collecting answers from more male respondents or dropping male respondents and making it clear already in the title that it's only about women. Let's hope that a different sample size will lead to a better Cronbach alpha. We invite you to submit a revised version of the manuscript that addresses the points raised during the review process. Please disregard the comment about APA citation style.

We look forward to receiving your revised manuscript.

Kind regards,

Frantisek Sudzina

Academic Editor

PLOS ONE

Journal Requirements:

2. In the online submission form, you indicated that “The data that support the findings of this study are available from the corresponding author, KS, upon reasonable request.”.

3. Please remove your figures from within your manuscript file, leaving only the individual TIFF/EPS image files, uploaded separately. These will be automatically included in the reviewers’ PDF.

Reviewers' comments:

Reviewer's Responses to Questions

**Comments to the Author**

1. Is the manuscript technically sound, and do the data support the conclusions?

Reviewer #1: Partly

Reviewer #2: Yes

2. Has the statistical analysis been performed appropriately and rigorously? 

Reviewer #1: No

Reviewer #2: Yes

3. Have the authors made all data underlying the findings in their manuscript fully available?

Reviewer #1: No

Reviewer #2: Yes

4. Is the manuscript presented in an intelligible fashion and written in standard English?

Reviewer #1: Yes

Reviewer #2: Yes

5. Review Comments to the Author

Reviewer #1: This is a cross-sectional study exploring how loneliness is related to delusion-like experiences in the general population, and whether procrastination and boredom proneness mediate this relationship. The sample includes 331 participants, and analyses are conducted using bootstrapped mediation models. Even if the paper is interesting, I found specific aspects that require careful attention by the authors:

- There are some problems with the participants. A specific gender imbalance, 82.8% of the sample are women. This limits generalizability and is not sufficiently discussed. Moreover, the use of self-selected online sample may not reflect the broader population in terms of education, socioeconomic status, etc.

- Low reliability of Boredom Proneness Scale: Cronbach's alpha = 0.52 (total score) is very low, questioning the validity of conclusions about boredom proneness.

- Procrastination and boredom proneness may be conceptually overlapping or collinear (as the manuscript itself notes), raising questions about their simultaneous inclusion in the model.

- While procrastination is said to “reduce” the effect of loneliness on delusions, the language implies a causal direction not warranted by the cross-sectional design.

- No data are presented on psychiatric history or current mental health. This limits the interpretability of results regarding non-clinical vs. clinical delusion-like experiences.

- The data is said to be available “upon reasonable request.” This does not meet PLOS ONE’s standard of open data unless a strong justification is provided.

Reviewer #2: Thanks to the author for their work in addressing many of my previous comments. In my opinion, the manuscript is much improved with these changes:

There remain some changes that I have proposed that they authors have not fully integrated into their manuscript, and I would encourage them to reconsider. I attempt to clarify and expand on my comments below.

1- What is novelty in this study? There are so many studies, systematic reviews and meta-analysis conducted on this domain.

2- This generalization regarding some results needs further clarification and evidence as it applies to Western society. To show critical involvement, briefly mention any methodological flaws in the referenced studies or discrepancies in the results. The sentences in the introduction are somewhat complex and could be simplified for better readability. There is also redundancy in some sections.

3-This sudden transition to hypothesis one, two, three - there must be evidence and a prelude to the hypothesis - through a simple introduction.

4- In the methodology part, it is best to create a table showing all the information for the first study in terms of age, gender, education, and all descriptive data. The method section needs more detail about participant recruitment, inclusion/exclusion criteria, and demographic details. For example; At the beginning of the explanation of the questionnaires and methodology, no details were mentioned about the sample size - they were mentioned later in the results- Regarding the sample number and age, we need a table showing the division of age into categories because it is considered a very important indicator of significance.

5- There are a lot of grammatical errors and repetition - please review your language and sentence structure well. Structure of content (not of headings), is not easy to follow. For example, the first point should lead to the second. Some sentences are too long; and the grammar and English language needs attention.

6-The discussion section is too brief and lacks depth. Engage more critically with the findings, particularly the lack of significant gender differences, and the implications of the study How do these findings contribute to the existing body of literature on delusion.

7-Review all of the reference according to the final version of APA ; for example, there is no DOI (Digital Object Identifier) for the reference. How can the results be integrated and commented on with future research, as well as what are the shortcomings of this research (restructure all of this with main subheadings for both the benefits and shortcomings, as well as future research).

6. PLOS authors have the option to publish the peer review history of their article (what does this mean? ). If published, this will include your full peer review and any attached files.

**Do you want your identity to be public for this peer review?** For information about this choice, including consent withdrawal, please see our Privacy Policy .

Reviewer #1: No

Reviewer #2: **Yes: ** Mai Helmy

---

## [Author Response · Author response to Decision Letter 1]

7 Jul 2025

The main change involved limiting the obtained results exclusively to women, which was highlighted by modifying the title, updating the theoretical section of the manuscript, and re-running all statistical analyses on the female subgroup. Additionally, the noted limitations regarding the data collection method and the low Cronbach’s alpha reliability coefficient for one of the instruments were appropriately justified in the paragraph describing the study’s limitations.

Response to Reviewer #1’s comments:

First, I would like to sincerely thank the reviewer for their valuable and constructive comments. The insights provided have been very helpful in improving the quality and clarity of the manuscript. I have carefully considered each point and addressed them as follows:

1) The sample was reduced exclusively to the group of women; the limitation of data collection via electronic means as a common method was justified in the study’s limitations section.

2) The reliability of the BPS instrument changed after reducing the sample size; moreover, this aspect was addressed in the limitations section with appropriate explanations.

3) Prior to conducting the mediation analysis, the variables were checked for multicollinearity criteria (VIF and Tolerance coefficients); the results did not indicate any multicollinearity issues. An explanation has been added in the “Results” section.

4) I am aware that this was not an experimental study, so causal relationships cannot be inferred. The direction of the relationships was based on theoretical and empirical assumptions presented in the theoretical section. Consequently, all conclusions were stated probabilistically. The penultimate paragraph of the discussion acknowledges the impossibility of claiming causality, which would require further experimental or longitudinal studies.

5) The lack of data on psychiatric history or current mental health status of participants was described in the study limitations.

6) The data have been made publicly available via the OSF platform (https://osf.io/smky9/?view_only=339a37dac63a490eb54ccc0424c7962d).

Response to Reviewer #2’s comments:

I would like to sincerely thank Reviewer #2 for thorough and insightful feedback. The constructive suggestions provided have greatly contributed to enhancing the manuscript. Below, I address each of the points raised:

1) The novelty of my study is the specific configuration of variables examined, which has not been previously investigated. In particular, the mediating role of procrastination and boredom proneness has not been studied in relation between loneliness and delusion-like experiences before. Moreover, this configuration has not been explored specifically in relation to women, which adds a unique perspective and contribution to the existing literature.

2) I have added a reference to a study that specifically addresses the experience of loneliness within both collectivistic and individualistic cultural contexts.

3) Under each hypothesis, paragraphs with theoretical assumptions and previous research findings that supported the hypotheses formulated in this article were included. In my opinion, this was done clearly.

4) I added a table with the demographic characteristics of the sample, and updated the description of the sample as well as the participant recruitment procedure in the methods section.

5) The text was edited and reviewed by a professional translation agency.

6) The study was limited exclusively to the group of women. The discussion section was updated accordingly.

7) DOI numbers were added to the missing references (some did not have identifiers). The sections on study limitations and future research directions were separated into distinct subsections.

---

## [Editor Report · Decision Letter 1]

PONE-D-25-06976R1Loneliness and delusion-like experiences among women: mediating role of procrastination and boredom pronenessPLOS ONE

Dear Dr. Szalińska,

Thank you for submitting your manuscript to PLOS ONE. After careful consideration, we feel that it has merit but does not fully meet PLOS ONE’s publication criteria as it currently stands. Therefore, we invite you to submit a revised version of the manuscript.

Please add composite reliabilities to strengthen your paper. Please remove the "author's summary".

We look forward to receiving your revised manuscript.

Kind regards,

Frantisek Sudzina

Academic Editor

PLOS ONE

Journal Requirements:

Additional Editor Comments:

Please add composite reliabilities to strengthen your paper. Please remove the "author's summary".

---

## [Author Response · Author response to Decision Letter 2]

18 Jul 2025

I appreciate all of the editor’s valuable comments and advices, which helped me to improve my paper. According to the last revision from the editor, I deleted the author summary section, add the composite reliability values to all of the instruments used in the study, and complete the discussion section respectively. I left the markings of the previous changes in the paper and add a new ones.

The main change involved limiting the obtained results exclusively to women, which was highlighted by modifying the title, updating the theoretical section of the manuscript, and re-running all statistical analyses on the female subgroup. Additionally, the noted limitations regarding the data collection method and the low Cronbach’s alpha reliability coefficient for one of the instruments were appropriately justified in the paragraph describing the study’s limitations.

Response to Reviewer #1’s comments:

First, I would like to sincerely thank the reviewer for their valuable and constructive comments. The insights provided have been very helpful in improving the quality and clarity of the manuscript. I have carefully considered each point and addressed them as follows:

1) The sample was reduced exclusively to the group of women; the limitation of data collection via electronic means as a common method was justified in the study’s limitations section.

2) The reliability of the BPS instrument changed after reducing the sample size; moreover, this aspect was addressed in the limitations section with appropriate explanations.

3) Prior to conducting the mediation analysis, the variables were checked for multicollinearity criteria (VIF and Tolerance coefficients); the results did not indicate any multicollinearity issues. An explanation has been added in the “Results” section.

4) I am aware that this was not an experimental study, so causal relationships cannot be inferred. The direction of the relationships was based on theoretical and empirical assumptions presented in the theoretical section. Consequently, all conclusions were stated probabilistically. The penultimate paragraph of the discussion acknowledges the impossibility of claiming causality, which would require further experimental or longitudinal studies.

5) The lack of data on psychiatric history or current mental health status of participants was described in the study limitations.

6) The data have been made publicly available via the OSF platform (https://osf.io/smky9/?view_only=339a37dac63a490eb54ccc0424c7962d).

Response to Reviewer #2’s comments:

I would like to sincerely thank Reviewer #2 for thorough and insightful feedback. The constructive suggestions provided have greatly contributed to enhancing the manuscript. Below, I address each of the points raised:

1) The novelty of my study is the specific configuration of variables examined, which has not been previously investigated. In particular, the mediating role of procrastination and boredom proneness has not been studied in relation between loneliness and delusion-like experiences before. Moreover, this configuration has not been explored specifically in relation to women, which adds a unique perspective and contribution to the existing literature.

2) I have added a reference to a study that specifically addresses the experience of loneliness within both collectivistic and individualistic cultural contexts.

3) Under each hypothesis, paragraphs with theoretical assumptions and previous research findings that supported the hypotheses formulated in this article were included. In my opinion, this was done clearly.

4) I added a table with the demographic characteristics of the sample, and updated the description of the sample as well as the participant recruitment procedure in the methods section.

5) The text was edited and reviewed by a professional translation agency.

6) The study was limited exclusively to the group of women. The discussion section was updated accordingly.

7) DOI numbers were added to the missing references (some did not have identifiers). The sections on study limitations and future research directions were separated into distinct subsections.

---

## [Editor Report · Decision Letter 2]

Loneliness and delusion-like experiences among women: mediating role of procrastination and boredom proneness

PONE-D-25-06976R2

Dear Dr. Szalińska,

We’re pleased to inform you that your manuscript has been judged scientifically suitable for publication and will be formally accepted for publication once it meets all outstanding technical requirements.

Kind regards,

Frantisek Sudzina

Academic Editor

PLOS ONE
---

## [Editor Report · Acceptance letter]

PONE-D-25-06976R2

PLOS ONE

Dear Dr. Szalińska,

I'm pleased to inform you that your manuscript has been deemed suitable for publication in PLOS ONE. Congratulations! Your manuscript is now being handed over to our production team.

Kind regards,

on behalf of

Dr. Frantisek Sudzina

Academic Editor

PLOS ONE